# Minerals and Antioxidant Micronutrients Levels and Clinical Outcome in Older Patients Hospitalized for COVID-19 during the First Wave of the Pandemic

**DOI:** 10.3390/nu15061516

**Published:** 2023-03-21

**Authors:** Clément Lahaye, François Parant, Julie Haesebaert, Karine Goldet, Lamia Bendim’red, Laetitia Henaff, Mitra Saadatian-Elahi, Philippe Vanhems, Charlotte Cuerq, Thomas Gilbert, Emilie Blond, Muriel Bost, Marc Bonnefoy

**Affiliations:** 1Department of Geriatric Medicine, Hôpital Gabriel Montpied, 63000 Clermont-Ferrand, France; 2Unité de Nutrition Humaine, Université Clermont Auvergne, INRAE, 63000 Clermont-Ferrand, France; 3Biology Center South, Hôpital Lyon Sud, 69310 Pierre-Bénite, France; 4Public Health Unit, Department of Clinical Research and Epidemiology, Groupement Hospitalier Est, 69002 Lyon, France; 5RESHAPE Research on Healthcare Performance Inserm U1290, Université Lyon 1, 69008 Lyon, France; 6Clinical Research Centre, Ageing, Brain, Fragility-Hôpital des Charpennes, 69100 Villeurbanne, France; 7Department of Hygiene, Epidemiology and Prevention, Hôpital Édouard Herriot, Hospices Civils de Lyon, 69003 Lyon, France; 8ICIR-International Center for Infectiology Research (Team PHE3ID), Claude Bernard Lyon 1 University, Inserm, U1111, CNRS, UMR5308, ENS Lyon, 46 Allée d’Italie, 69007 Lyon, France; 9Department of Geriatric Medicine, Groupement Hospitalier Sud, CHU de Lyon, 69495 Pierre-Bénite, France; 10INSERM, 1060 CaRMeN 165 Chemin du Grand Revoyet, 69310 Pierre-Bénite, France

**Keywords:** minerals antioxidant micronutrients, clinical outcome, older patients, COVID-19, first wave of the pandemic

## Abstract

Excessive inflammatory response has been implicated in severe respiratory forms of coronavirus disease 2019 (COVID-19). Trace elements such as zinc, selenium, and copper are known to modulate inflammation and immunity. This study aimed to assess the relationships between antioxidant vitamins and mineral trace elements levels as well as COVID-19 severity in older adults hospitalized. In this observational retrospective cohort study, the levels of zinc, selenium, copper, vitamin A, β-carotene, and vitamin E were measured in 94 patients within the first 15 days of hospitalization. The outcomes were in-hospital mortality secondary to COVID-19 or severe COVID-19. A logistic regression analysis was conducted to test whether the levels of vitamins and minerals were independently associated with severity. In this cohort (average age of 78 years), severe forms (46%) were associated with lower zinc (*p* = 0.012) and β-carotene (*p* < 0.001) concentrations, and in-hospital mortality (15%) was associated with lower zinc (*p* = 0.009), selenium (*p* = 0.014), vitamin A (*p* = 0.001), and β-carotene (*p* = 0.002) concentrations. In regression analysis, severe forms remained independently associated with lower zinc (aOR 2.13, *p* = 0.018) concentrations, and death was associated with lower vitamin A (aOR = 0.165, *p* = 0.021) concentrations. Low plasma concentrations of zinc and vitamin A were associated with poor prognosis in older people hospitalized with COVID-19.

## 1. Introduction

The rapid spread across the world of a new coronavirus from the first quarter of 2020 has overwhelmed many healthcare systems, even in developed countries. This virus, transmitted largely by respiratory droplets and to a lesser extent by mucosal contact, named SARS-CoV-2 (severe acute respiratory syndrome coronavirus-2) because of the frequency of its severe respiratory forms, can also lead to digestive (anorexia, diarrhea…), vascular (heart attack…) and neurological (stroke, delirium…) symptoms [1]. These forms leading to acute respiratory distress syndrome requiring high-flow oxygen therapy or mechanical ventilation were associated with major mortality in the absence of specific treatment during the first wave of the disease, between March and June 2020 in Europe [2]. Thus, in 2020, coronavirus disease 2019 (COVID-19) has become the third leading cause of death behind cardiovascular pathologies and cancers in many countries such as in the United States or in France [3,4]. Age over 65 years, male gender, and comorbidities are major risk factors for severe forms and death during COVID-19 infection [5,6].

The susceptibility of the elderly to severe forms of COVID-19 was demonstrated from the beginning of the pandemic and relies notably on the presence of pathologies such as diabetes, hypertension, or obesity [7]. The particular criticality of COVID-19 in the elderly also implies immuno-senescence, which is a progressive process favored by comorbidities and certain nutritional deficits, leading both to a reduction in responses to new pathogens and also to excessive inflammatory responses [8,9]. Thus, cytokine storm, an uncontrolled secretion of chemokines and pro-inflammatory cytokines, can lead to multiple organ failure particularly respiratory with acute respiratory distress syndrome (ARDS), and it was recognized as the leading cause of death during the first wave of COVID [10,11]. Moreover, every stage of the immune response, both humoral and cellular, is dependent on nutritional factors [12]. Beyond protein-energy malnutrition, levels of various micronutrients are associated with the immune response in the event of COVID-19 [13,14]. Thus, micronutrients with antioxidant properties such as the metal and trace elements (zinc, iron, selenium and copper) and vitamins A, C, E, D, B (B6, B9, B12) and C are known to support adequate immune response [15,16]. For example, vitamin A contributes to the protection of mucosal and epithelium integrity, and it has regulatory functions on humoral and cellular immune responses, while zinc is a major factor in the growth, development, and maintenance of immune responses, especially against viral infections such as COVID-1- [17,18,19]. At the same time, insufficient intake and altered status of these nutrients concern a large proportion of the Western population, in particular the elderly [20]. In the absence of a vaccine or specific treatment during the first wave, certain anti-oxidant micronutrients were thus quickly proposed to optimize immunity against COVID-19 [21].

However, apart from vitamin D, studies involving the associations between plasma antioxidant or mineral micronutrients levels and the clinical outcomes of COVID-19 infection are still scarce [22,23]. We conducted a retrospective cohort study during the first wave of the pandemic to document the possible relationship between antioxidant levels and COVID-19 severity in older adults over 50 years hospitalized for COVID-19. The objectives of our study were (i) to identify associations between plasma minerals and antioxidant micronutrients levels at hospital admission and COVID-19 severe forms defined by the presence of one of the following criteria: high oxygen requirement, intubation, or death; and death related to COVID-19 during the hospital stay, (ii) to identify associations between plasma minerals and antioxidant micronutrients levels and age or inflammation level measured by C-reactive protein (CRP).

## 2. Materials and Methods

### 2.1. Study Design and Population Characteristics

The present study was a sub-study of the observational study “MicroCovAging” (micronutrients status in aging patients with COVID-19) conducted at the Hospices Civils de Lyon (HCL), in Lyon, France, and it focused on the potential link between plasma antioxidant micronutrients levels and COVID-19 severity. Micronutrients analysis was conducted on frozen tube bottoms (plasma heparinized samples) coming from routine sampling during COVID-19 patient hospitalizations, thanks to the COVID-19 biobank hosted at HCL Biological Resource Centre. Thus, the micronutrient assays as well as the routine assays (albumin, CRP in particular) were carried out on the same day.

Two hundred and thirty patients aged ≥50 years and hospitalized for active SARS-CoV-2 infection proven by Real-Time Polymerase Chain Reaction (RT-PCR) in the Lyon public hospitals were consecutively included during the first wave of the pandemic (admission date from 1 March 2020 to 30 June 2020). The patients were therefore affected by the Wuhan-Hu-1 strain of the SARS-CoV-2 and had not been exposed to any specific vaccine. Two patients were excluded from the data analysis due to a withdrawal of consent (see flow chart in Figure 1).

### 2.2. Ethical Consideration

The study was approved by the scientific and ethical committee of the HCL and was reported on the clinicaltrials.gov website (NCT04877509). Medical doctors informed patients of the study during hospitalization or by post, thanks to a generic information sheet dedicated to COVID-19 research at the HCL. All patients received an information note accompanied by a non-objection form and had one month to express their opposition. The processing of personal data carried out for this study falls within the scope of “Reference Methodology No. 4” (MR-004) of the Commission Nationale de l’Informatique et des Libertés (CNIL).

### 2.3. Data Collection

The NOSO-COR project, a prospective anonymous inpatient COVID-19 database also carried by the HCL, gave us access to patient demographics, clinical features and treatments [24]. Biochemical biomarkers (CRP, albumin, vitamin D) were obtained by routine testing during hospitalization.

### 2.4. Micronutrients Levels

Among plasma micronutrients, 6 were selected with regard to their potential importance based on knowledge from previous pathophysiological studies: minerals zinc, selenium, copper; and vitamins A, β-carotene, and E [25,26,27,28,29]. Only assays of the 94 patients whose samples have been collected during the first 15 days of hospitalization were considered for the analysis of the associations between plasma antioxidant or mineral micronutrients levels at hospital admission and COVID-19 severe forms and death. We have chosen to focus on the first 15 days of hospitalization in order to limit the biases linked on the one hand to phenomena of hyper-consumption possibly linked to the severity and duration of the disease and on the other hand to possible supplements after this period. Regarding the secondary objective (associations between plasma antioxidant micronutrients levels and age or inflammation level measured by CRP levels), the analyses were first carried out on the 94 patients and then extended to the population of 228 patients.

Levels of selenium were determined by Inductively Coupled Plasma–Mass Spectrometry (ICP–MS) on an Elan DRCe instrument (PerkinElmer) in Dynamic Reaction Cell (DRC) mode using oxygen as the reaction gas and with ^78^Se as the isotope of choice. The internal standard was rhodium. Inductively Coupled Plasma–Optical Emission Spectrometry (ICP–OES) (5110 instrument, Agilent, Santa Clara, CA, USA) was used to determine the levels of plasma copper and zinc. The emission wavelengths were 213.857 nm and 327.395 nm, respectively.

Vitamin A, vitamin E and β-carotene concentrations were measured by high-performance liquid chromatography coupled to spectrophotometric detection with an H-Class Waters chromatographic system (Waters, St Quentin-en-Yvelines, France). Before analysis by chromatography, vitamin A, vitamin E, and β-carotene were extracted from plasma heparinized samples conserved at <−18 °C. Briefly, for the extraction, plasma was mixed during 10 min with internal standard and ethanol and then during 30 min with water and hexane. After centrifugation during 10 min at 5300× *g*, the upper organic layer was evaporated under nitrogen flux. Dry residue was dissolved in methanol/ethanol mix (85/15, *v*/*v*) and injected in the chromatographic system. Separation was performed on a kinetex 2.6 µ polar C18 (Phenomenex, Le Pecq, France) using a gradient elution with acetonitrile/methanol/isopropanol (78/20/2, *v*/*v*/*v*) as solvent A and acetonitrile/methanol/isopropanol (40/40/20, *v*/*v*/*v*) as solvent B for a total run of 13 min. The gradient was as follows: 0 min (100% A and 0% B), 2 min (0% A and 100% B), and 8.5 min (100% A and 0% B) with a constant flow of 0.2 mL/min. The wavelengths used for vitamin A, vitamin E, and β-carotene determination were 325 nm, 292 nm and 450 nm, respectively. Quantification was performed by using Empower3_HF1_Enterprise software (version 7.30.00.00, Waters). The laboratory regularly participates in external audits attesting to the reliability of the measurements of trace elements and vitamins.

### 2.5. Study Clinical Endpoints of COVID-19 Disease Severity

The clinical endpoints of the study were: (i) the in-hospital mortality directly related to COVID-19 and (ii) the onset of severe forms of COVID-19. Two independent physicians carried out the assignment of causes of death via a medical evaluation. The severe forms of COVID-19 were defined as any of the following criteria: high oxygen requirement (i.e., oxygen flow > 5 L/min), intubation, or death related to COVID-19 during the hospital stay. ICU admissions as such were not included as a severe form criterion because older patients were frequently ineligible for ICU admission (age over 80 years, or Rockwood Clinical Frailty scale greater than 5, or patient refusal of an ICU transfer) [30].

### 2.6. Statistical Analysis

Since the quantitative variables distribution significantly deviated from a normal distribution (*p* < 0.001) (except for body mass index (BMI) and albumin), we expressed results as medians and interquartile ranges (IQR) and used a non-parametric test for all comparisons (Mann–Whitney U-test). The test used for data distribution to test for normality was the Kolmogorov–Smirnov test. Logistic regression models were performed to separately explain each clinical endpoint (death or severe forms) according to a series of variables considered in the literature as risk factors for a severe form of COVID-19 (age, BMI, gender, CRP and albumin). Thus, logistic regression analysis took into account: age (≤75 vs. >70 years old), gender, BMI (<35 kg/m^2^/≥35 kg/m^2^), CRP and albumin levels. As we have previously shown within the same cohort that vitamin D supplementation taken during the 3 months preceding the infection onset may have a protective effect on the development of severe COVID-19 forms in older adults, pre-hospital vitamin D supplementation (yes/no) was also considered [23]. For continuous covariates, the results of the logistic regression analysis are traditionally presented as the odds ratio reflecting the increase in risk of an event occurring for a one unit increase in the continuous variable. To improve clinical relevance and facilitate interpretation, depending on the micronutrient studied, we calculated the OR for a more clinically relevant increase or decrease than one unit. Thus, the legend of the table shows for which variation of the dosage the odds ratio has been calculated. All tests were two-tailed, and the significance level was defined when *p* value < 0.05. Statistical analyses were performed using SAS 9.4 software (2013, SAS Institute Inc., Cary, NC, USA).

## 3. Results

### 3.1. Patient Demographics and Comorbidities

Table 1 shows the baseline characteristics of the 228 enrolled patients hospitalized with SARS-CoV-2 infection confirmed by RT-PCR. The median age was 78 years (IQR, 68–87), with a male predominance (56%). Twenty-five percent of patients lived in an institution before their hospitalization. Multiple comorbidities were present in 63% of the patients: 56% had arterial hypertension, 36% had other cardiovascular diseases, 32% had diabetes mellitus (types 1 and 2), 19% had neuro-cognitive disorders, 14% had renal diseases, 13% had pulmonary diseases, 12% had cancer and 7% had a BMI greater than 35 kg/m^2^.

The full enrolled population in the “MicroCovAging” study has similar characteristics to those of the sub-population of patients (n = 94) whose samples for micronutrients analysis have been collected during the first 15 days (median: 8 days; IQR: 5–12 days) of hospitalization (Fisher’s exact tests non-significant).

### 3.2. Clinical Outcomes

Among the 228 patients, 105 (46%) developed a severe form of COVID-19, 84 (37%) were admitted to the ICU and 35 (15%) deaths related to COVID-19 occurred during hospitalization (Table 1). Sixteen deceased patients (16/35) died before ICU admission or were ineligible for ICU admission.

Age of 70 years and over was associated with COVID-19 mortality (91% vs. 72%, *p* = 0.004). Male patients were overrepresented among severe forms of COVID-19 (sex ratio = 2.2, *p* < 0.001), admission to ICU (sex ratio = 2.5, *p* < 0.001) and COVID-19 mortality (sex ratio = 4.0, *p* = 0.003). Patients with grade 2 and 3 obesity (BMI ≥ 35 kg/m^2^) were overrepresented in severe forms (12% vs. 3%, *p* = 0.02).

### 3.3. Association between Plasma Antioxidant and Mineral Micronutrients Levels and Clinical Outcomes

Among the 94 patients with micronutrients analysis collected during the first 15 days of hospitalization, severe forms were associated with lower zinc (0.55 vs. 0.66 mg/L, *p* = 0.012), β-carotene (0.28 vs. 0.42 µmol/L, *p* < 0.001) concentrations; and death with lower zinc (0.51 vs. 0.62 mg/L; *p* = 0.009), selenium (53 vs. 65 µg/L, *p* = 0.014), vitamin A (0.79 vs. 1.38 µmol/L, *p* = 0.001), β-carotene (0.24 vs. 0.37 µmol/L, *p* = 0.002), and vitamin E concentrations (23 vs. 27 µmol/L, *p* = 0.047) (Table 2 and Table 3).

In multivariate analysis, severe forms remained independently associated with lower zinc (aOR 2.13, *p* = 0.018) concentrations; death was associated with lower vitamin A (aOR = 0.165, *p* = 0.021) concentrations (Table 4). Whatever the micronutrient considered, the increase in albumin was accompanied by a reduction in the risk of a severe form without a significant effect of age or BMI (see Appendix A). However, age > 75 years and BMI ≥ 35 kg/m^2^ were the factors regularly associated with mortality.

### 3.4. Association between Plasma Antioxidant Micronutrients Levels and CRP Levels or Age

Among the 94 patients, a significant negative relationship was observed with CRP for zinc (r = −0.591, *p* < 0.001), selenium (r = −0.357, *p* = 0.002), vitamin A (r = −0.561, *p* < 0.001), β-carotene (r = −0.397, *p* < 0.001), and vitamin E (r = −0.245, *p* = 0.019); conversely, a positive relationship was observed for copper (r = 0.362, *p* < 0.001). Increasing age was associated with lower zinc (r = −0.234, *p* = 0.023) and selenium (r = −0.281, *p* = 0.014), but it was non-significant with vitamin A, vitamin E and β-carotene concentrations. These results were similar to those obtained by extending the analysis to the total population of 228 patients: a significant negative relationship was observed with CRP for zinc (r = −0.338, *p* < 0.001), selenium (r = −0.352, *p* < 0.0001), vitamin A (r = −0.543, *p* < 0.001), β-carotene (r = −0.421, *p* < 0.001), and vitamin E (r = −0.173, *p* = 0.01); conversely, a positive relationship was observed for copper (r= 0.368, *p* < 0.001). Increasing age was associated with lower zinc (r = −0.234, *p* = 0.023), selenium (r = −0.302, *p* < 0.001), vitamin A (r = 0.217, *p* < 0.001), and vitamin E (*p* = −0.193, *p* = 0.003) but higher β-carotene (r = 0.194, *p* = 0.03) concentrations.

## 4. Discussion

In this observational study, we have shown an association between low plasma zinc level (measured within 15 days of admission) and the risk of severe form as well as an association between low plasma level of vitamin A and mortality related to COVID-19 independently of well-recognized risk factors such as age, sex or obesity. In addition, age above 70 was associated with lower plasma levels of zinc, selenium, vitamin A and vitamin E. Finally, plasma zinc, selenium, vitamin A, β-carotene and vitamin A levels were inversely proportional to CRP levels.

Our results are consistent with those of the literature concerning a link between prognosis in respiratory infections and plasma levels of certain micronutrients, vitamin A and zinc being frequently cited. In a representative population of 34,000 participants, lower plasma levels of vitamins A, C, D, and α-tocopherol vitamin E were associated with increased respiratory morbidity and/or mortality in U.S. adults [31]. In a 9-year longitudinal study, low levels of selenium were associated with shorter survival in the elderly [32]. Low plasma levels of zinc and vitamin A have been associated with poor prognosis in COVID-19 with an increased risk of ICU admission, longer disease duration or higher mortality [33,34,35]. In a population of 34 patients hospitalized for COVID-19, a deficiency in selenium, vitamin D, vitamin A, and zinc was present in 51%, 40%, 39%, and 39%, respectively [36]. Moreover, a lower risk for severe COVID-19 was associated with higher levels of vitamin A (aOR 0.18, 95% CI 0.05–0.69, *p* = 0.01), zinc (aOR 0.73, 95% CI 0.55–0.98, *p* = 0.03), and folic acid (aOR 0.88, 95% CI 0.78–0.98, *p* = 0.02). Low zinc levels (<0.5 mg/L) at admission have been associated with more severe clinical presentation and higher mortality in SARS-CoV-2 Infection [34]. Selenium deficiency seems particularly frequent in patients hospitalized for COVID-19, which can concern up to 42% of them [37]. In another cohort of COVID-19, non-survivors were characterized by an overrepresentation of a simultaneous zinc and selenium transporter selenoprotein *p* (SELENOP) deficiency [33]. A Spanish observational study found that patients with low levels of zinc or vitamin A more often required ICU admission during the first wave of the pandemic and that low levels of vitamin A were associated with the risk of orotracheal intubation [35]. However, the results are divergent, and some studies have not found a link between selenium and survival during COVID-19 [38].

One major hypothesis may be that micronutrients deficiency results in altered immune responses and thus in an impaired secretion of cytokines and antibody response leading to impaired defenses against viral infections [39]. Low selenium status has been also associated with poor musculoskeletal function, which can promote respiratory dysfunction [40]. Zinc is a cofactor of >300 enzymes and plays a critical role in both innate and adaptive immune responses [41]. Zinc deficiency may also trigger elevating inflammatory response by promoting aberrant immune cell activation [42]. Vitamin A plays a regulatory role in both innate and adaptive (cellular and humoral) immunity [17]. Indirectly, micronutrient deficiencies can also promote bacterial superinfections (respiratory but also urinary), which are frequently observed during hospitalization for COVID-19, and they may aggravate respiratory damages or increase mortality [43].

Among patients admitted to an ICU for COVID-19, there was 20% of the over 70-year-olds and only 5% of the over 80-year-olds. This underrepresentation of the elderly in the ICU fundamentally differs from the profile of patients hospitalized with COVID-19 and reflects the exclusion from the ICU of the oldest and comorbid patients favored by insufficient ICU capacities during the first wave as well as by their ability to recover [44].

In our study, age over 70 was associated with lower micronutrient levels, which is consistent with the literature. In a systematic review in 2020, Vural et al. found significant proportions (25 to 50%) of populations either community-based or institutionalized, showing insufficiency for zinc, selenium, iron, iodine and copper [45]. Insufficient intake related to both quantitative and qualitative alteration of the diet in the older population may promote low levels of micronutrients together with protein-energy malnutrition in this population [46]. The fact that zinc and vitamin A remain factors independently associated with severe forms and mortality in our study, even after adjustment for albuminemia, shows that the dosage of certain micronutrients could also provide additional prognostic information to protein-energetic malnutrition.

Our results have shown an inverse correlation between the level of inflammation measured by CRP and the plasma level of most micronutrients. This relationship can be explained in both directions: either by an excess of consumption in case of a severe inflammatory response or by a pre-existing deficit influencing the inflammatory response. In a prospective study of critically ill adults, Koekkoek et al. have shown that 24 ICU patients presented with lower plasma levels of selenium, β-carotene, vitamin C, and vitamin E than 21 age-matched healthy controls [47]. Moreover, immune dysfunction and systemic inflammation can be promoted by zinc deficiency [42]. The low level of micronutrients such selenium or zinc, observed in COVID-19 populations compared to control populations, does not necessarily imply pre-existing deficiencies favoring the occurrence of the disease [48,49]. As has been shown in the treatment of certain cancers, it is also possible that COVID-19 itself leads to an alteration in micronutrient status [50]. For example, the combination of hypoxia and IL-6 suppresses selenoprotein expression (such as glutathione peroxidase), causing whole-body selenium status decline and an increased generation of reactive oxygen species [51]. More generally, redistribution, altered protein binding, increased losses (through urine, blood, sweat, ascites, pleural fluid), increased metabolic use, and dilution secondary to fluid resuscitation may explain low micronutrient levels in ICU [52]. More likely, the low levels of micronutrients observed in critically ill patients could be the result of a variable combination of pre-existing deficiencies, disease-related or treatment-related perturbations [53].

This study also raises the issue of supplementation. Interventions based on micronutrient supplementation in the general population have so far shown unconvincing effects. Some reviews plead for supplementation with micronutrients as a safe, effective, and low-cost strategy to help support optimal immune function, especially against acute respiratory tract infections [54,55]. These outcomes differ from other studies where the supplementation of ICU patients (mostly sepsis or ARDS) with antioxidant micronutrients demonstrate neutral outcomes and even harm [52]. The support in micronutrients goes above all through a balanced diet with regular intake of fruits and vegetables and complete cereals even also for some of them by supplements (such as vitamin D) [56,57]. Selenium and zinc supplementation may improve immune response in critically ill patients with acute respiratory distress syndrome related to COVID-19 [58]. Interestingly, in a recent meta-analysis, zinc supplementation in COVID-19 appeared to be associated with a trend toward reduced mortality (RR 0.79, 95% CI 0.60–1.03, *p* = 0.08) [59].

### Limitations and Strengths of the Study

Our study has some limitations. First, the level of evidence of our study is limited by its observational retrospective nature, but the brutal occurrence of the pandemic did not give us the possibility of adopting a more demanding design. Secondly, we only demonstrated a significant and independent association with the prognosis for two of the six micronutrients considered, which may be partly linked to low power related to the limited number of patients, but reinforces the relevance of the results obtained. The design of the study and the limited clinical data collected did not allow us to analyze other factors such as secondary infections or organ decompensation (cardiac, renal, etc.), which can themselves be the cause of an excess of morbidity or mortality and potentially affect micronutrient or CRP levels [60]. Moreover, some patients, especially those treated in the ICU, could have been supplemented with micronutrients, although there were no recommendations at that time. The retrospective design inherent to the sudden onset of the first wave of the COVID-19 epidemic did not allow us to collect this information in an exhaustive manner. Nevertheless, the doses used always correspond to the recommended daily doses and were therefore unlikely to affect plasma levels in such a short time.

However, this study has several strengths. This is one of the few studies to focus on the impact of micronutrients on the prognosis of COVID-19 disease, while most of the studies have essentially considered protein-energy malnutrition. In addition, this study included older to very old populations and therefore, it is particularly representative of the populations affected by severe forms of COVID-19. Moreover, the richness of the data collected allowed us to identify an association between various micronutrients and prognosis, which is truly independent of many confounding factors such as age, sex, albumin or obesity.

## 5. Conclusions

Low plasma concentrations of zinc and vitamin A were associated with poor prognosis in elderly people hospitalized with COVID-19 during the first wave of the pandemic. These results, in agreement with other observational studies in the literature, emphasize that the nutritional challenges of COVID-19 involve both micro and macronutrient issues. The simultaneous analysis of a panel of micronutrients reinforces the interest of our study.

Although literature data remain rare concerning the elderly population and insufficient to distinguish between insufficient intake and increased consumption in the acute phase, these results may plead for the specific supplementation of elderly patients in order to strengthen their anti-viral defenses, limit cytokine storm and improve prognosis.

## Figures and Tables

**Figure 1 nutrients-15-01516-f001:**
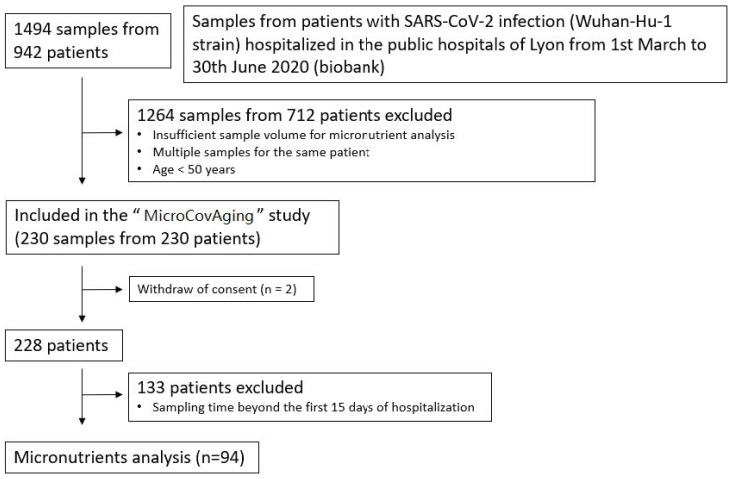
Flow chart of the patients included in the study.

**Table 1 nutrients-15-01516-t001:** Baseline demographics, comorbidities, biomarkers, and clinical outcomes of the 228 patients, including the 94 patients with micronutrients analysis collected during the first 15 days of hospitalization.

Parameters	Total Patients	COVID-19 Survivors	COVID-19- Related Death	Severe COVID-19 (Including Death) *	Non-Severe COVID-19	Primary Objective Subgroup
	(n = 228)	85% (193/228)	15% (35/228)	46% (105/228)	54% (123/228)	(n = 94)
Median age (IQR)	78 years (68–87)	76 years (66–87)	82 years (76–86)	73 years (66–82)	82 years (71–90)	79 years (70–88)
Age ≥ 80 years % (n)	43% (99/228)	41% (80/193)	54% (19/35)	27% (28/105)	58% (71/123)	47% (44/94)
Sex ratio (male/female)	1.3 (129/99)	1.1 (101/92)	4.0 (28/7)	2.2 (72/33)	0.9 (57/66)	0.9 (45/49)
Living in institutions % (n)	25% (56/228)	24% (47/193)	26% (9/35)	19% (20/105)	29% (36/123)	38% (36/94)
Median BMI (IQR)	25.7 kg/m^2^ (22.1–29.1) n = 224 **	25.9 kg/m^2^ (22.2–29.1) n = 190	24.7 kg/m^2^ (20.6–27.6) n = 34	26.7 kg/m^2^ (23.8–30.4) n = 104	24.3 kg/m^2^ (21.3–28.6) n = 120	24.5 kg/m^2^ (21.1–28.6) n = 94
BMI ≥ 35 kg/m^2^ % (n)	7% (16/224)	6% (11/190)	15% (5/34)	12% (12/104)	3% (4/120)	10% (9/94)
Median CRP (IQR)	40 mg/L (8–105) (n = 220)	29 mg/L (6–78) (n = 186)	102 mg/L (54–181) (n = 34)	179 mg/L (111–271) (n = 104)	20 mg/L (5–64) (n = 121)	55 mg/L (24–124) (n = 91)
Median albumin (IQR)	27.3 g/L (23.0–32.4) (n = 214)	28.3 g/L (24.2–33.3) (n = 183)	22.0 g/L (18.7–26.1) (n = 31)	19.7 g/L (16–23.9) (n = 98)	30.0 g/L (26.7–34.1) (n = 117)	27.3 g/L (23.5–31.6) (n = 86)

Abbreviations: BMI, body mass index, COVID-19, coronavirus disease 2019; CRP, C-reactive protein; IQR, interquartile range; * One of the following criteria: high oxygen requirement, intubation, or death related to COVID-19 during the hospital stay. ** BMI was not available in 4 patients.

**Table 2 nutrients-15-01516-t002:** Plasma antioxidant micronutrients levels and COVID-19 severity among the 94 patients with micronutrients analysis collected during the first 15 days of hospitalization.

Antioxidant Micronutrients	Non-Severe COVID-19 (n = 53)	Severe COVID-19, Including Death (n = 41)	*p*-Value * (z Value)	COVID-19 Survivors (n = 78)	COVID-19- Related Death (n = 16)	*p*-Value * (z Value)
Zinc (mg/L)	0.66 (0.53–0.76) (n = 53)	0.55 (049–0.68) (n = 41)	**0.012** (2.505)	0.62 (0.52–0.75) (n = 78)	0.51 (0.44–0.59) (n = 16)	**0.009** (2.591)
Selenium (µg/L)	64 (49–78) (n = 43)	62 (45–74) (n = 33)	NS (0.650)	65 (50–77) (n = 63)	53 (39–61) (n = 13)	**0.014** (2.456)
Copper (mg/L)	1.20 (1.02–1.41) (n = 53)	1.20 (1.0–1.4) (n = 41)	NS (0.335)	1.21 (1.03–1.41) (n = 78)	1.13 (0.99–1.34) (n = 16)	NS (0.805)
Vitamin A (µmol/L)	1.35 (0.95–1.81) (n = 53)	1.15 (0.74–1.71) (n = 41)	NS (1.647)	1.38 (0.93–1.88) (n = 78)	0.79 (0.61–1.00) (n = 16)	**0.001** (3.229)
β-carotene (µmol/L)	0.42 (0.29–0.67) (n = 53)	0.28 (0.19–0.38) (n = 41)	**0.001** (3.477)	0.37 (0.26–0.57) (n = 78)	0.24 (0.14–0.29) (n = 16)	**0.002** (3.029)
Vitamin E (µmol/L)	25.7 (21.5–29.9) (n = 53)	26.2 (21.2–29.9) (n = 41)	NS (0.175)	26.6 (21.5–30.9) (n = 78)	23.0 (18.9–26.5) (n = 16)	**0.047** (1.982)

Plasma levels determined during the first 15 days of hospitalization Abbreviations: IQR, interquartile range. * Mann–Whitney U-test. Values of *p* < 0.05 are shown in bold.

**Table 3 nutrients-15-01516-t003:** Univariate logistic regression analysis of the association between plasma micronutrients levels and COVID-19-related deaths and severe forms in the 94 patients.

	COVID-19-Related Deaths			Severe COVID-19		
Parameters	Crude OR	95% CI	*p*-Value	Crude OR	95% CI	*p*-Value
Age > 75 years	3.348	0.883–12.694	0.0755	0.595	0.258–1.375	0.2246
Male sex	2.847	0.903–8.973	0.0740	1.510	0.665–3.428	0.3242
CRP (mg/L) *	1.004	0.998–1.010	0.1739	1.005	1.000–1.011	0.0614
Vit D supp	0.768	0.243–2.432	0.6536	0.583	0.245–1.387	0.2225
Albumin (g/L) **	0.918	0.825–1.022	0.1183	0.741	0.646–0.850	**<0.0001**
BMI ≥ 35 (kg/m^2^)	2.875	0.632–13.085	0.1720	2.765	0.646–11.838	0.1706
Zinc (mg/L) ^#^	1.683	1.076–2.631	**0.0225**	1.411	1.057–1.884	**0.0196**
Selenium (µg/L) ^###^	1.727	1.110–2.687	**0.0154**	1.120	0.890–1.410	0.3345
Copper (mg/L) ^#^	1.038	0.864–1.246	0.6926	0.956	0.836–1.093	0.5085
Vitamin A (µmol/L) ^#^	1.214	1.060–1.390	**0.0051**	1.060	0.992–1.132	0.0840
β-carotene (µmol/L) ^#^	1.455	1.038–2.040	**0.0294**	1.421	1.142–1.767	**0.0016**
Vitamin E (µmol/L) ^##^	1.011	0.965–1.060	0.6393	1.011	0.965–1.060	0.6393

Abbreviations: OR, odds ratio; Vit D supp, pre-hospital vitamin D supplementation. * OR for the increase of 1 mg/L, ** OR for the increase of 1 g/L, ^#^ OR for the decrease of 0.1 unit, ^##^ OR for the decrease of 1 units, ^###^ OR for the decrease of 10 units. Values of *p* < 0.05 are shown in bold.

**Table 4 nutrients-15-01516-t004:** Multivariate logistic regression analysis of the association between plasma micronutrients levels and COVID-19-related deaths and severe forms (adjusted on age, sex, CRP, albumin, BMI and vitamin D supplementation) in the 94 patients.

	COVID-19-Related Deaths			Severe COVID-19		
Antioxidant Micronutrients	Adjusted OR	95% CI	*p*-Value	Adjusted OR	95% CI	*p*-Value
Zinc (mg/L) ^#^	1.396	0.778–2.506	0.2631	2.133	1.139–3.993	**0.0179**
Selenium (µg/L) ^###^	1.506	0.880–2.579	0.1356	1.157	0.752–1.780	0.5065
Copper (mg/L) ^#^	1.025	0.815–1.289	0.8334	0.963	0.774–1.198	0.7341
Vitamin A (µmol/L) ^#^	1.298	1.041–1.620	**0.0206**	1.022	0.919–1.137	0.4025
β-carotene (µmol/L) ^#^	1.243	0.825–1.874	0.2985	1.230	0.904–1.672	0.1871
Vitamin E (µmol/L) ^##^	1.043	0.942–1.155	0.4135	1.004	0.934–1.079	0.5873

^#^ OR for the decrease of 0.1 unit, ^##^ OR for the decrease of 1 unit, ^###^ OR for the decrease of 10 units. Values of *p* < 0.05 are shown in bold.

## Data Availability

The data presented in this study are available on request from the corresponding author. The data are not publicly available due to ethical considerations, regarding personal information and respecting what was written in the generic information sheet dedicated to COVID-19 research at the Hospices Civils de Lyon (Lyon, France).

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
