# Peer review of "Minerals and Antioxidant Micronutrients Levels and Clinical Outcome in Older Patients Hospitalized for COVID-19 during the First Wave of the Pandemic"

_nutrients, 2023, doi:10.3390/nu15061516_

Round 1

Reviewer 1 Report

Please see my suggestions below:

7.5 pages of real manuscript for 12 authors.

On the manuscript, the type of the paper (Article, Review, etc.) is not specified, above the title. Please complete (check the nutrients draft for manuscript)

Keywords must reflect the main characteristic words of the paper (usually reflected also by the title) in the best way to increase the paper's relevance and chances to be find when searching it after key words.  So, for the actual title, I suggest the following relevant keywordsminerals; antioxidant micronutrients; clinical outcome; older patients; COVID-19; first wave of the pandemic.

Set the entire paper in Palatino linotype.

Please check the Instructions for authors regarding references insertion in the main text – in brackets (not in parenthesis) and revise the entire manuscript accordingly.

In the entire manuscript please revise Abbreviations as the Instructions for authors must be checked and respected “Acronyms/Abbreviations/Initialisms should be defined the first time they appear in each of three sections: the abstract; the main text; under the first figure or table. When defined for the first time, the acronym/abbreviation/initialism should be added in parentheses after the written-out form”. Revise the entire manuscript in this regard.

Introduction section is much too short. Please improve it. In this regard, I suggest checking and referring to https://doi.org/10.3389/fcell.2020.00616https://doi.org/10.1155/2022/1571826 and https://doi.org/10.1007/s11356-022-22345-w

The 2nd section is again numbered as 1. Please correct in L75.

Instead Figure 1, please provide a CONSORT type flow chart.

Remove the empty lines in the main text. They give a careless appearance to your paper.

L160. Kg/sq m or Kg/m2 not m2. Conceptually is incorrect.

Tables 1 and 2. Head of the table. Please be completed also for the first column (Parameters? for the Table 1, and Antioxidant micronutrients? for the Table 2). Same observation for Tables 3 and 4.

What is the meaning of the bolded p-value in the Tables 3 and 4?

Also, in the Tables 3 and 4, I suggest changing $ symbol (which is confusing) with #, and adding it as Superscript.

In the statistical part:

- the test used for data distribution is not specified. It is only mentioned that the data are not normally distributed. It is advisable to specify the test as well.

-           Table 1; in the row with the median, below the median in parenthesis, I assume it is the interval from the minimum to the maximum, but it can also be the 95% confidence interval...; it is not specified what the row with parenthesis means. Practically not a parenthesis is explained...; what do the elements in parentheses mean? if you work with min and max, or with average plus minus 2* standard deviation

- in addition to p values, specify the t value (Statistical value) (also for the chi-square test, mention the test value, not just the p values).

Conclusions section. Please underline the scientific relevance of your results.

References section. Please check the Instructions for authors regarding setting and MDPI style of references and apply.

Author Response

RESPONSES TO REVIEWERS

Minerals and antioxidant micronutrients levels and clinical outcome in older patients hospitalized for COVID-19 during the first wave of the pandemic

We thank the reviewers for all these remarks allowing us to further improve the quality of the text.

You will find our point-to-point answers in the attached document and a version of the word with track changes is attached to the revision.

Reviewer 1

7.5 pages of real manuscript for 12 authors.

On the manuscript, the type of the paper (Article, Review, etc.) is not specified, above the title. Please complete (check the nutrients draft for manuscript).

> This number of authors reflects the mobilization of staff to participate in original studies during the COVID period. Moreover, it is a sub-study of a larger database project which has mobilized many personnel (methodologists, biologists and clinicians).

Keywords must reflect the main characteristic words of the paper (usually reflected also by the title) in the best way to increase the paper's relevance and chances to be found when searching it after key words.  So, for the actual title, I suggest the following relevant keywords: minerals; antioxidant micronutrients; clinical outcome; older patients; COVID-19; first wave of the pandemic.

> Thank you for this relevant remark which will improve our visibility during bibliographical research. The keywords have been changed accordingly.

Set the entire paper in Palatino linotype.

> This change has been made.

Please check the Instructions for authors regarding references insertion in the main text – in brackets (not in parenthesis) and revise the entire manuscript accordingly.

> This change has been made in the text and the presentation of the references.

In the entire manuscript please revise Abbreviations as the Instructions for authors must be checked and respected “Acronyms/Abbreviations/Initialisms should be defined the first time they appear in each of three sections: the abstract; the main text; under the first figure or table. When defined for the first time, the acronym/abbreviation/initialism should be added in parentheses after the written-out form”. Revise the entire manuscript in this regard.

> The entire manuscript has been revised accordingly.

Introduction section is much too short. Please improve it. In this regard, I suggest checking and referring to https://doi.org/10.3389/fcell.2020.00616, https://doi.org/10.1155/2022/1571826 and https://doi.org/10.1007/s11356-022-22345-w

> Indeed, the introduction has been extensively revised and enriched, using  in particular on these references, in order to better describe both the clinical manifestations of the disease and the possible links with micronutrients.

The 2nd section is again numbered as 1. Please correct in L75.

>  Indeed, the change has been made.

Instead Figure 1, please provide a CONSORT type flow chart.

> Figure 1 has been modified to match the CONSORT format. Some clarifications have been added in the materials and methods section.

Remove the empty lines in the main text. They give a careless appearance to your paper.

> The entire manuscript has been revised accordingly.

L160. Kg/sq m or Kg/m2 not m2. Conceptually is incorrect.

>  Indeed, the change has been made.

Tables 1 and 2. Head of the table. Please be completed also for the first column (Parameters? for the Table 1, and Antioxidant micronutrients? for the Table 2). Same observation for Tables 3 and 4.

> Indeed, the tables have been completed.

What is the meaning of the bolded p-value in the Tables 3 and 4?

>  Values of p < 0.05 are shown in bold. this precision has been added to the legend of the tables.

Also, in the Tables 3 and 4, I suggest changing $ symbol (which is confusing) with #, and adding it as Superscript.

>  Indeed, these changes have been made.

In the statistical part:

- the test used for data distribution is not specified. It is only mentioned that the data are not normally distributed. It is advisable to specify the test as well.

> The test used for data distribution to test for normality was the Kolmogorov-Smirnov test. This element is now specified in the methods.

- Table 1; in the row with the median, below the median in parenthesis, I assume it is the interval from the minimum to the maximum, but it can also be the 95% confidence interval...; it is not specified what the row with parenthesis means. Practically not a parenthesis is explained...; what do the elements in parentheses mean? if you work with min and max, or with average plus minus 2* standard deviation

> This point is indeed imprecise. Age, BMI, CRP, and albumin are presented as median value and interquartile range. This point has been clarified in table 1.

- in addition to p values, specify the t value (Statistical value) (also for the chi-square test, mention the test value, not just the p values).

> We did not perform, but only Mann-Whitney tests in table 2 and logistic regressions in tables 3 and 4. We have therefore added the values of the Mann-Whitney U tests (z-value) in table 2.

Conclusions section. Please underline the scientific relevance of your results.

> The conclusion has been revised to highlight the scientific relevance of our work.

References section. Please check the Instructions for authors regarding setting and MDPI style of references and apply.

> This change has been made in the text and the presentation of the references.

Reviewer 2 Report

The presented study is a substudy which examined if minerals and antioxidant micronutrients have an impact on the outcome in older patients hospitalzided for C19.  They found that low plasma conc. of zinc and vitamin A were associated with poor prognosis in elderly hospitalized covid patients. 

The manuscript and study is well written and of interest for all clinicians.

However i feel some points should be adressed before the manuscript is ready to be published;

-What was the rational of excluding the non severe C19 patients in the regression analysis? 

-The authors should add a statement, or correct for superinfections. Many severe COVID patients get secondarly pneumonia or infection of the lower urinary tract. How homo/heterogenous were the cohort in regards to comorbidity and complications of other infectious diseases? I guess a secondary infection can alter the measured values as well.

I think the authors can adress my few points in a timely manner.

-

Author Response

RESPONSES TO REVIEWERS

Minerals and antioxidant micronutrients levels and clinical outcome in older patients hospitalized for COVID-19 during the first wave of the pandemic

We thank the reviewers for all these remarks allowing us to further improve the quality of the text.

You will find our point-to-point answers in the attached document and a version of the word with track changes is attached to the revision.

Reviewer 2

The presented study is a substudy which examined if minerals and antioxidant micronutrients have an impact on the outcome in older patients hospitalzided for C19.  They found that low plasma conc. of zinc and vitamin A were associated with poor prognosis in elderly hospitalized covid patients. 

The manuscript and study is well written and of interest for all clinicians.

However i feel some points should be adressed before the manuscript is ready to be published;

-What was the rational of excluding the non severe C19 patients in the regression analysis? 

> Indeed, some formulations of the method paragraph are unclear. Severe COVID-19 were not excluded from the regression analysis. The primary objective (associations between plasma antioxidant or mineral micronutrients levels at hospital admission and COVID-19 severe forms and death) was analyzed on a sub-population (n=94) Regarding the secondary objective (associations between plasma antioxidant micronutrients levels and age or inflammation level measured by CRP levels), the analyzes were first carried out on the 94 patients and then extended to the population of 228 patients. Changes in the methods and results paragraphs have been made to make these elements clearer.

-The authors should add a statement, or correct for superinfections. Many severe COVID patients get secondarly pneumonia or infection of the lower urinary tract. How homo/heterogenous were the cohort in regards to comorbidity and complications of other infectious diseases? I guess a secondary infection can alter the measured values as well.

 > Indeed, we considered the severe forms according to the respiratory criterion. Nevertheless, the design of the study and the limited clinical data collected did not allow us to analyze other factors such as secondary infections (respiratory, urinary, etc.) or organ decompensation (cardiac, renal, etc.), which can themselves be the cause of morbidity or mortality and potentially affect micronutrient or CRP levels. This observation was specified in the discussion paragraph.

Reviewer 3 Report

This is an interesting manuscript. The authors are kindly invited to revisit the following points.

1. Please revisit the phrasing of the abstract for grammatical errors that affect the meaning and try to keep the same format throughout the text for abbreviations and micronutrients.

2. Line 48: Please consider using the following format: SARS-CoV-2.

3. Line 49: Please revisit phrasing and include relevant references.

4. Line 53 (and throughout the text): Please try to keep the same format for the references (before punctuation), as indicated by the journal’s guidelines.

5. Lines 53 and 59: Please consider potential misunderstandings if both COVID and COVID-19 are present in the text. The relevant references cited by the authors seem to reflect COVID-19 and not coronaviruses in general. Kindly revisit and revise properly.

6. Line 65: Please include relevant references to support this statement.

7. Line 74: Kindly consider including the full word for CRP before using the abbreviated.

8. a) Introduction - Kindly revisit the text for some unnecessary spaces and punctuation that might have escaped the authors while typing.

8. b) Introduction - Also, it might be beneficial to the reader if the authors could include in the introduction a brief description of what is considered a severe form of COVID-19 since it is presented as one of the study’s main targets.

9. Line 184-185: Please revisit the phrasing and also kindly consider including the information of what might have been the criteria for ICU admission that some patients seem to have failed to meet.

10. Line 184: Revisit the reference to Table 1 for a typo.

11. Table 1: Please revisit the BMI for a typo.

12. Table 1: Patients living in institutions – I seem to have missed where this information is presented in the text before its reference in the Table. Kindly consider its inclusion and discussion if appropriate.

13. Line 189: The obesity footnote seems to be missing here.

14. Table 1 and Table 2: Kindly consider including the baseline information for the population sample that was evaluated for Plasma antioxidant micronutrients. As it is the reader has a view of the overall sample baseline and then the measurements refer to a sub-group of that without a clear understanding of “who these people were”. Perhaps it would be beneficial to elaborate.

15. Line 206: BMI ≥ 35 kg/m² tends to be associated with severe forms – Please consider elaborating on how is this supported by the outcomes presented.

16. Line 243: Kindly consider including relevant references to support this statement. Also, it might be beneficial to elaborate on what is considered a poor prognosis (ICU admission? Longer hospitalization? Death?)

17. 243-251: Please revisit for typos. Also kindly consider discussing the relevant previous studies in relation to this work as it may be helpful for the readers to have a better view of these relations rather than presenting them with previous outcomes.

18. Line 300-301: Kindly consider rephrasing. What may be considered critically ill?

19. Given the particular difficulties of all the studies conducted during the early stages of the pandemic, limitations in the study design should not deter its presence in the literature. However, the authors may want to consider a more thorough introduction and discussion.  

Author Response

RESPONSES TO REVIEWERS

Minerals and antioxidant micronutrients levels and clinical outcome in older patients hospitalized for COVID-19 during the first wave of the pandemic

We thank the reviewers for all these remarks allowing us to further improve the quality of the text.

You will find our point-to-point answers in the attached document and a version of the word with track changes is attached to the revision.

Reviewer 3

This is an interesting manuscript. The authors are kindly invited to revisit the following points.

  1. Please revisit the phrasing of the abstract for grammatical errors that affect the meaning and try to keep the same format throughout the text for abbreviations and micronutrients.

> The entire manuscript has been revised accordingly.

  1. Line 48: Please consider using the following format: SARS-CoV-2.

> Indeed, this change has been made.

  1. Line 49: Please revisit phrasing and include relevant references.

> This sentence, as well as large parts of the introduction, has been reformulated and some references have been added. 

  1. Line 53 (and throughout the text): Please try to keep the same format for the references (before punctuation), as indicated by the journal’s guidelines.

> The entire manuscript has been revised accordingly.

  1. Lines 53 and 59: Please consider potential misunderstandings if both COVID and COVID-19 are present in the text. The relevant references cited by the authors seem to reflect COVID-19 and not coronaviruses in general. Kindly revisit and revise properly.

> Indeed, all references relate to COVID-19. The entire manuscript has been revised accordingly.

  1. Line 65: Please include relevant references to support this statement.

> A reference has been added.

Gombart, A.F.; Pierre, A.; Maggini, S. A Review of Micronutrients and the Immune System–Working in Harmony to Reduce the Risk of Infection. Nutrients 2020, 12, 236, doi:10.3390/nu12010236.

  1. Line 74: Kindly consider including the full word for CRP before using the abbreviated.

> Indeed, this abbreviation has been defined.

  1. a) Introduction - Kindly revisit the text for some unnecessary spaces and punctuation that might have escaped the authors while typing.

> The entire manuscript has been revised accordingly.

  1. b) Introduction - Also, it might be beneficial to the reader if the authors could include in the introduction a brief description of what is considered a severe form of COVID-19 since it is presented as one of the study’s main targets.

> This clarification was added at the end of the introduction.

“The objectives of our study were i) to identify associations between plasma antioxidant or mineral micronutrients levels at hospital admission and COVID-19 severe forms (any of the following criteria: high oxygen requirement, intubation, or death) and death related to COVID-19 during the hospital stay and mortality, ii) to identify associations between plasma antioxidant micronutrients levels and age or inflammation level measured by C-reactive protein (CRP).”

  1. Line 184-185: Please revisit the phrasing and also kindly consider including the information of what might have been the criteria for ICU admission that some patients seem to have failed to meet.

> These were the oldest patients (over 80 years old), those with at least moderate frailty (Rockwood Clinical Frailty scale greater than 5), or those refusing a transfer to ICU. This clarification has been added to the methods. ICU admissions as such were not considered as a severe form criterion because patients were frequently ineligible for ICU admission (age over 80 years, or Rockwood Clinical Frailty scale greater than 5, or patient refusal of an ICU transfer). »

Because of these sometimes subjective criteria, admission to the ICU was not analyzed in the primary endpoint, but patients requiring high-flow oxygen therapy were indeed included in the severe forms, which was one of the major reasons for admission during the first wave.

It was a typo.

  1. Line 184: Revisit the reference to Table 1 for a typo.

> The correction has been made.

  1. Table 1: Please revisit the BMI for a typo.

> The correction has been made.

  1. Table 1: Patients living in institutions – I seem to have missed where this information is presented in the text before its reference in the Table. Kindly consider its inclusion and discussion if appropriate.

> Indeed, this information has been added. “25% of patients lived in an institution before their hospitalization.”

  1. Line 189: The obesity footnote seems to be missing here.

> It was a typo. The definition of grade 2-3 obesity (BMI ≥ 35 kg/m²) has been added.

  1. Table 1 and Table 2: Kindly consider including the baseline information for the population sample that was evaluated for Plasma antioxidant micronutrients. As it is the reader has a view of the overall sample baseline and then the measurements refer to a sub-group of that without a clear understanding of “who these people were”. Perhaps it would be beneficial to elaborate.

> Indeed, table 2 refers to the sub-population of patients whose samples for micronutrients analysis have been collected during the first 15 days of hospitalization. The characteristics of these 94 patients have been specified in Table 1.

  1. Line 206: BMI ≥ 35 kg/m² tends to be associated with severe forms – Please consider elaborating on how is this supported by the outcomes presented.

> This assertion is exaggerated, the sentence has been reformulated: « Whatever the micronutrient considered, the increase in albumin was accompanied by a reduction in the risk of a severe form, whereas a BMI ≥ 35 kg/m² tends to be associated with severe forms, without a significant effect of age or BMI (Supplementary Table 1). »

A Supplementary Table has been added concerning multivariate analysis at the end of the manuscript.

  1. Line 243: Kindly consider including relevant references to support this statement. Also, it might be beneficial to elaborate on what is considered a poor prognosis (ICU admission? Longer hospitalization? Death?)

> Some references have been added and poor prognosis was defined. “Low micronutrient values have been associated with poorer prognosis in COVID-19 (ICU admission, disease duration, mortality) [26–28].”

  1. 243-251: Please revisit for typos. Also kindly consider discussing the relevant previous studies in relation to this work as it may be helpful for the readers to have a better view of these relations rather than presenting them with previous outcomes.

> This part of the discussion has been edited to better position our results in relation to the literature. Many typographical errors have been corrected.

  1. Line 300-301: Kindly consider rephrasing. What may be considered critically ill?

> This sentence refers to ICU patients hospitalized for sepsis or ARDS. This sentence has been rephrased. « These outcomes differ from other studies where supplementation of ICU patients (mostly sepsis or ARDS) with antioxidant micronutrients demonstrate neutral outcomes and even harm.”

  1. Given the particular difficulties of all the studies conducted during the early stages of the pandemic, limitations in the study design should not deter its presence in the literature. However, the authors may want to consider a more thorough introduction and discussion.  

> Introduction and discussion sections have been enriched.

Round 2

Reviewer 1 Report

The authors significantly improved the manuscript based on the suggestions received.

Author Response

I am very honored that our modifications gave you satisfaction. Thanks again for your expert advice.

Reviewer 2 Report

The authors adressed all my comments sufficiently. I think the manuscript is now ready to be published.

Author Response

(The authors gave the same response as above.)

Reviewer 3 Report

The authors responded to all suggestions.

Please be sure to include proper references of all Tables within the text.

Author Response

I am very honored that our modifications gave you satisfaction.

References to tables in the text have been checked. However, for greater consistency, the paragraph referring to the multivariate analysis, originally located before table 3, has been moved between tables 3 and 4.

Thanks again for your expert advice.